# Clinical Decision Support Systems for Antibiotic Prescribing: An Inventory of Current French Language Tools

**DOI:** 10.3390/antibiotics11030384

**Published:** 2022-03-14

**Authors:** Claire Durand, Serge Alfandari, Guillaume Béraud, Rosy Tsopra, François-Xavier Lescure, Nathan Peiffer-Smadja

**Affiliations:** 1Infection Antimicrobials Modelling Evolution (IAME) UMR 1137, University of Paris, French Institute for Medical Research (INSERM), F-75018 Paris, France; xavier.lescure@aphp.fr (F.-X.L.); nathan.peiffer-smadja@aphp.fr (N.P.-S.); 2Infectious Disease Department, University Hospital of Nice, F-06202 Nice, France; 3Infectious Disease Department, Gustave Dron Hospital, F-59200 Tourcoing, France; alfandari.s@gmail.com; 4Infectious Disease Department, University Hospital of Poitiers, F-86000 Poitiers, France; beraudguillaume@gmail.com; 5Information Sciences to Support Personalized Medicine, Centre de Recherche Des Cordeliers, Université de Paris, Sorbonne Université, INSERM, F-75006 Paris, France; rosy.tsopra@aphp.fr; 6HeKA, Inria Paris, F-75012 Paris, France; 7Department of Medical Informatics, Hôpital Européen Georges-Pompidou, Assistance Publique-Hôpitaux de Paris, F-75015 Paris, France; 8Infectious Disease Department, Bichat-Claude Bernard Hospital, Assistance-Publique Hôpitaux de Paris, F-75018 Paris, France; 9National Institute for Health Research Health Protection Research Unit in Healthcare Associated Infections and Antimicrobial Resistance, Imperial College London, London W12 0NN, UK

**Keywords:** antimicrobials, antibiotic prescribing, antimicrobial stewardship, clinical decision support system, CDSS

## Abstract

Clinical decision support systems (CDSSs) are increasingly being used by clinicians to support antibiotic decision making in infection management. However, coexisting CDSSs often target different types of physicians, infectious situations, and patient profiles. The objective of this study was to perform an up-to-date inventory of French language CDSSs currently used in community and hospital settings for antimicrobial prescribing and to describe their main characteristics. A literature search, a search among smartphone application stores, and an open discussion with antimicrobial stewardship (AMS) experts were conducted in order to identify available French language CDSSs. Any clinical decision support tool that provides a personalized recommendation based on a clinical situation and/or a patient was included. Eleven CDSSs were identified through the search strategy. Of the 11 CDSSs, only 2 had been the subject of published studies, while 9 CDSSs were identified through smartphone application stores and expert knowledge. The majority of CDSSs were available free of charge (*n* = 8/11, 73%). Most CDSSs were accessible via smartphone applications (*n* = 9/11, 82%) and online websites (*n* = 8/11, 73%). Recommendations for antibiotic prescribing in urinary tract infections, upper and lower respiratory tract infections, and digestive tract infections were provided by over 90% of the CDSSs. More than 90% of the CDSSs displayed recommendations for antibiotic selection, prioritization, dosage, duration, route of administration, and alternative antibiotics in case of allergy. Information about antibiotic side effects, prescription recommendations for specific patient profiles and adaptation to local epidemiology were often missing or incomplete. There is a significant but heterogeneous offer for antibiotic prescribing decision support in French language. Standardized evaluation of these systems is needed to assess their impact on antimicrobial prescribing and antimicrobial resistance.

## 1. Introduction

Antimicrobial resistance (AMR) is a major public health concern worldwide [1,2]. It is associated with high morbidity and mortality as well as significant healthcare costs [2]. In response to the global threat of AMR, antimicrobial stewardship programs (ASPs) have been introduced to optimize antibiotic use and to improve the quality of infection care [3,4]. ASPs have been proven to be effective to tackle AMR in hospital and community settings [5,6]. Moreover, ASPs based on physician education and increased availability of guidelines through decision support tools such as clinical decision support systems (CDSSs) have shown significant results in improving appropriate antibiotic prescribing [7,8].

CDSSs are computerized tools designed to support diagnostic or therapeutic decision-making in order to improve clinical practice and quality of care [9,10]. Upon providing information about a given clinical context and patient characteristics, clinicians are offered easy and quick access to up-to-date clinical practice guidelines (CPGs) at the point of care [9,10]. In the infectious diseases (ID) field, CDSSs have been increasingly used to assist physicians’ decision-making in antibiotic management in both community and hospital settings [11,12,13,14]. With a few clicks, CDSSs provide expert or evidence-based recommendations to promote the appropriate choice of antibiotics, dosage, route of administration, and duration of treatment.

One of the first CDSSs that was developed in medicine was MYCIN [15]. MYCIN was an expert system designed in the 1970s for both the diagnosis and treatment of infectious diseases [15]. Then with the emergence of evidence-based medicine in the 1990s [16], new CDSSs have been developed with the purpose of implementing CPGs. CDSSs have since shown many benefits such as improvement in antibiotic selection [17,18], reduction in antibiotic usage [19,20,21,22], reduction in broad-spectrum antibiotic use [22,23], shorter length of hospital stay [17,19], reduction in adverse events [19,20], decreased mortality [20], increase in pharmacy interventions [19], and decreased healthcare costs [17,19,21].

A systematic review has been performed on studies assessing CDSSs for antimicrobial management but was limited by publication bias and targeted a broad range of different clinical tasks, such as alert systems for pharmacists or tools for antimicrobial stewardship (AMS) teams to review prescriptions [13]. Moreover, newer systems are now available including innovative tools [24,25] and applications available on smartphones.

Therefore, this study aims to provide an up-to-date inventory of French language CDSSs that are currently used in community and hospital settings by expert and non-expert physicians. The purpose of this study was to describe existing CDSS, including those not cited in the scientific literature, and to provide their main characteristics and usage data.

## 2. Material and Methods

### 2.1. Search Strategy

A literature search was carried out in February 2021 to identify published articles about the design, the implementation or the evaluation of French language CDSSs for antimicrobial prescribing. The Pubmed/MEDLINE database was searched using MeSH terms and text words for antimicrobials and CDSSs, including synonyms. The Pubmed search strategy can be found in the Appendix A (Appendix A). Additional search terms such as “France” and “French” were included in the search strategy to restrict the search to French-language CDSSs. The reference lists of related reviews and systematic reviews were also searched to identify any relevant study that might have been missed by the search strategy.

Additionally, an open discussion was conducted with AMS experts from the Antimicrobial Stewardship study group of the French Infectious Diseases Society to identify CDSSs that are used in common practice by French-speaking primary care and hospital physicians, including those that have not been the subject of published research. They were asked to report any CDSS that can support physicians in community or hospital settings in the prescription of empirical or targeted antimicrobial therapy. 

We also searched smartphone application stores such as App Store (iOs) and Play Store (Android) using French keywords such as “antibiotique” (French word for antibiotic), “antibiothérapie” (French word for antibiotic therapy) or “prescription” (French word for prescription).

### 2.2. CDSS Selection

Any French-language clinical decision support tool that provides a personalized recommendation based on a clinical situation and/or a patient was included. Electronic tools available on smartphone applications, stand-alone software, and online websites were all included. Tools that exclusively provide a list of official practice guidelines or information about a single clinical situation were not included. Applications or websites that only offer teleconsultation services, drug monographs, or veterinary prescribing guidelines were also excluded.

### 2.3. Data Collection

A data collection form was developed and reviewed by ID specialists and AMS physicians using 5 randomly selected CDSSs. Collected data included the CDSS characteristics regarding their administration, access, targeted healthcare providers and patients, search criteria, types of infection, and types of information provided.

After identifying available decision support tools, testing was carried out by two researchers using the standardized form. Testing was performed after installation on a server in order to have reproductible testing procedure. Data from the included CDSSs were recorded by two reviewers independently and were then subjected to further critical appraisal during a narrative synthesis.

## 3. Results

Figure 1 describes the CDSS selection process that was undertaken. Through the Pubmed search strategy, we identified and screened 35 articles. After assessment of eligibility and exclusion of duplicates, only 2 CDSSs were included in the inventory from the literature search [24,25]. Seven other CDSSs were identified and included in the inventory through the open discussion with AMS experts. Two additional CDSSs were found by the search on application stores, after excluding one CDSS intended for antibiotic prescribing in veterinary medicine. A total of 11 CDSSs were thus included in the inventory: Antibioclic, Antibiogarde, Antibiogilar, antibioGUIDE (Perpignan), Antibioguide (Basse-Normandie), AntibioEst, APPLIBIOTIC, ePOPI, Prescriptor, Antibiothérapie Pédiatrique, AntibioHelp^®^.

Included CDSSs were then tested using the standardized form. The collected data are described in Table 1 and detailed for each CDSS in the Appendix A. One CDSS was unavailable for testing so we contacted its main administrator to obtain its characteristics.

Out of the 11 decision support systems included, 10 CDSSs were designed by French AMS teams whereas 1 CDSS was developed by Canadian physicians and was intended for pediatrics use only. Most of the CDSSs were less than 10 years old and were developed on a regional scale by multidisciplinary teams including general practitioners (GPs), ID specialists, emergency physicians, intensive care physicians, pediatricians, geriatricians, microbiologists, pharmacists, and medical informatics specialists. Nine and eight support systems were accessible respectively via smartphone applications and online websites. Two of the CDSSs available on smartphone applications could only be accessed through one or another mobile operating system, Android or iOS. Eight CDSSs could be used offline on smartphone. Moreover, 8 CDSSs were available free of charge. 

The individual characteristics of each CDSS regarding targeted users, patients, and infections are presented in Table 2. All the CDSSs offered prescription recommendations for the ambulatory treatment of community-acquired infections. In addition, all CDSSs except two (Antibioclic and AntibioHelp^®^) were also intended for the treatment of inpatients in hospital settings. Urinary tract infections (UTIs) were the only type of infection for which all decision support tools provided prescription recommendations. Furthermore, UTIs, upper respiratory tract infections (URTIs), lower respiratory tract infections (LRTIs), and digestive tract infections were the only types of infection for which over 90% of the decision support systems provided recommendations. In contrast, recommendations for the treatment of cardiovascular infections, bloodstream infections, central venous catheter (CVC) related infections, eye infections, and dental infections were the least frequently advised among the CDSSs with less than half of the CDSSs providing treatment recommendations for these conditions.

Table 3 describes the individual characteristics of the included CDSSs regarding the types of information provided. All the decision support systems provided recommendations for the decision to initiate antibiotic therapy for a given infection, the selection of appropriate antibiotics as well as their preferred order according to guidelines. All included tools also displayed alternatives in case of allergy. All but one of the CDSSs also provided decision support on the appropriate dosage and duration of treatment. However, less than 30% of the CDSSs displayed information about antibiotic side effects. Two CDSSs required clinicians to systematically provide the patient profile (i.e., adult, child, pregnant woman) prior to displaying prescription recommendation. Recommendations for specific patient profiles such as children or pregnant women were frequently provided by the CDSSs but the information supplied was often incomplete. Recommendations for antibiotic selection and dosage in patients with chronic kidney disease (CKD) were displayed by about half of the CDSSs. Moreover, less than 30% of the CDSSs displayed antibiotic prescription recommendations adapted to the local epidemiology. In addition to displaying prescription recommendations, the majority of CDSSs displayed additional information about infection epidemiology, clinical presentation, diagnosis, and other treatment. Nine CDSSs displayed the sources of their recommendations, including primarily national and international guidelines from scientific societies. 

## 4. Discussion

This study provides an overview of available French language CDSSs and their characteristics. Although this inventory might not be exhaustive, our main objective was to identify and to describe the CDSSs that are used by clinicians for antibiotic prescribing. To the best of our knowledge, there is no published research describing and comparing CDSS for antibiotic prescribing in a similar way. We found that two CDSSs (Antibioclic and AntibioHelp^®^) were particularly suitable for use in primary care settings. Indeed, these two CDSSs focused on the infectious situations most frequently encountered in general and emergency medicine and displayed comprehensive prescription recommendations for different patient profiles (i.e., adults, children, pregnant women). One CDSS (ePOPI) met all the predefined criteria regarding targeted users, patients, infectious situations, and recommendations, although it should be noted that this CDSS was not free of charge. Another CDSS (Antibiothérapie Pédiatrique) focused solely on pediatrics and offered comprehensive recommendations for a range of infectious situations in this area. Two other CDSSs (APPLIBIOTIC and AntibioEst) targeted a variety of infections in both general and specialized medicine and thus seemed appropriate for decision support in both inpatient and outpatient settings. It is reasonable to infer from these results that the appropriateness of a CDSS for a physician likely depends on his or her scope of practice and patient profile.

Despite the growing use of CDSSs, only 2 of the 11 CDSSs included in the inventory appear to have been the subject of published studies. This lack of published research on existing tools highlights currents gaps in the evaluation of CDSSs and their potential impact on antibiotic prescribing behavior and clinical outcomes. In a study published in 2020 [26], Delory et al. described the architecture of Antibioclic and its use. They reported its growing number of users and queries as well of the nature of these queries, including mostly URTIs and UTIs. They also reported the findings of two cross-sectional studies conducted five years apart with Antibioclic users through an online survey [25]. Among the 1848 and 3621 survey participants, 93% were physicians and 81% were GPs. The vast majority of GPs (93%) reported following the CDSS recommendations while the occurrence of CDSS users’ non-compliance with the decision to initiate an antibiotic, select an antibiotic and extend the duration of treatment beyond the CDSS recommendation decreased between the two surveys. A substantial number of GPs declared using the CDSS to update their knowledge on antibiotic therapy with a decrease over time between the two surveys (83% in 2014 versus 43% in 2019), suggesting an increase in user knowledge of antibiotic prescribing guidelines over time. However, the authors reported that no formal assessment of the CDSS impact on improving antibiotic prescribing practices has been carried out. Another CDSS included in this inventory has been the subject of small-scale evaluation [25]. AntibioHelp^®^ aims to help GPs extrapolate guideline recommendations to clinical situations and patients for which there are no explicit recommendations [25]. By displaying antibiotic properties weighted by degree of importance in addition to displaying recommended and non-recommended antibiotics according to guidelines, this CDSS promoted a better understanding of recommendations and encouraged the weighing of pros and cons of each antibiotic in decision making by clinicians. The use of AntibioHelp^®^ by GPs resulted in a significant improvement in antibiotic prescribing in situations when no explicit recommendation existed [25] as well a significant increase in GP confidence in guideline recommendations [25]. The provision of flexible and comprehensible recommendations therefore appears to be an important factor to consider to increase the uptake of CDSSs by clinicians. Indeed, several studies have reported a correlation between CDSS adoption and their positive impact on antibiotic prescribing [11,12,13], which highlights the need to assess not only the effects of CDSSs on antibiotic prescribing but also their implementation process and utilization. Given the link between the uptake of CDSSs by clinicians and their effectiveness on improving antibiotic prescribing behavior, it is crucial to understand the limits of CDSSs and the characteristics that influence clinician adoption, to allow for the development of new research methods to overcome these limits. In order to optimize current CDSSs and to improve their sustainability, current gaps in the evaluation of CDSS utilization, user satisfaction, and impact on clinician adherence to guidelines should be addressed by CDSS administrators.

Despite the paucity of CDSSs described in the scientific literature, we found a significant offer for French language CDSSs showing a strong interest from multidisciplinary physician teams to improve antibiotic use. Included CDSSs were simple to use, with most support systems merely requiring users to provide the site and nature of infections, making them easy to use by non-expert physicians. Most support systems were found to offer prescription recommendations for a variety of infectious diseases, making them valuable to different types of physicians, both general and specialist, and useful in both primary care and hospital settings. We found that UTIs, URTIs, LRTIs, and digestive tract infections were the infectious situations for which the most CDSSs provided recommendations whereas bloodstream infections and CVC-related infections were advised by only a few CDSSs. This may reflect the priority given to the most frequent indications for antibiotic prescribing or the most frequent causes of antibiotic misuse. This also highlights the fact that CDSSs are probably easier to develop for the treatment of simple community-acquired infections, given their narrower spectrum of causative pathogens and infrequent multidrug resistance. Indeed, guidance for the treatment of healthcare-associated infections requires in most cases more detailed information about patient history, clinical presentation, previous antibiotic exposure, and proper examination of microbiological test results. The development of knowledge-based CDSSs for these difficult clinical situations would hence require a large volume of rules to capture expert knowledge. To this day, the use of CDSSs for antibiotic management seems more appropriate in general and emergency medicine practice or for the inpatient treatment of simple community-acquired infections. In contrast, therapeutic decision-making for the management of severe infections in hospital settings requires individualized expert guidance and follow-up from hospital AMS teams.

Three CDSSs, namely Antibiogarde, ePOPI, and Antibiothérapie Pédiatrique, were not free of charge and were available on an annual subscription basis, ranging from 7.99 to 33 EUR per year. Although fee-based access may significantly limit the uptake of these CDSSs given free coexisting options, it is worth mentioning that all three of these CDSSs had specific features, including prescribing guidelines for fungal and parasitic infections, guidelines for the diagnosis of infections and comprehensive guidelines for the management of neonatal and pediatric infections. Recommendation updates were infrequent in some decision support systems and recommendations for specific patient profiles such as children, pregnant women, or patients with CKD were missing or incomplete in some decision support systems. CDSSs’ lack of explicit recommendations for some clinical situations and populations is likely a barrier to widespread adoption by clinicians and could potentially contribute to delayed or inappropriate prescribing in these situations. Moreover, we found that information on the use and administration of CDSSs was sometimes missing or incomplete. Overall greater transparency could also promote better prescriber adherence to decision support systems. It is also worth noting that many of the included CDSSs appear to overlap and provide the same type of recommendations for the same patient profiles. Therefore, it may be worthwhile to find ways to centralize the process of computerizing antibiotic therapy recommendations in order to pool the resources invested in the development and sustainability of CDSSs, whether for antimicrobial prescribing or other clinical decisions.

All the study support systems were accessible through online websites, stand-alone applications or computer software and were not integrated into the electronic health record (EHR) workflow, which means every request on these CDSSs has to be activated by clinicians. The development of automated clinical decision support delivered through EHRs may increase user adoption. Furthermore, all the CDSSs presented in this study were knowledge-based systems, i.e., they provide recommendations based on expert medical knowledge. None of them used machine learning algorithms to recognize patterns in clinical data in order to predict patient outcomes, which is likely related to the lack of clinical data warehouses in primary care [27]. One narrative review has investigated the use of machine learning decision support systems (ML-CDSSs) in infectious diseases and found only three ML-CDSSs intended for decision making in antibiotic therapy selection while most existing ML-CDSSs focused on the diagnosis of infection and the prediction, early detection or stratification of sepsis [27]. Combining expert knowledge and machine learning algorithms could allow for personalized and predictive recommendations tailored to patient profiles and could thus have a positive impact on the quality of antibiotic prescribing. All CDSSs identified in this article were intended for medical prescribers. However, other healthcare providers such as pharmacists have been playing a growing role in ASPs [28,29,30] and could potentially rely on CDSSs for reviewing antibiotic prescriptions [31,32]. In addition, a few CDSSs offered the possibility to be parameterized locally to be adapted to the local epidemiology which could further optimize antibiotic prescribing and positively impact local antimicrobial resistance patterns.

## 5. Conclusions

This inventory shows a significant but heterogeneous offer for antibiotic prescribing decision support. Based on these results, a physician’s choice of CDSS should presumably depend on his or her scope of practice and patient profile. Most CDSSs provided recommendations for a range of infections, although few CDSSs offered comprehensive recommendations for antibiotic prescribing in specific patient profiles, which may limit adoption by clinicians. Frequent updates, free use, comprehensive recommendations, and automated clinical decision support are important factors to consider to increase the uptake of CDSSs by clinicians and thus their effectiveness in improving the quality of antibiotic prescribing and clinician adherence to guidelines. Moreover, findings from this study highlight current gaps in the evaluation of CDSS use and impact on antimicrobial prescribing and antimicrobial resistance. Standardized evaluation of current CDSSs is needed to optimize current tools and to improve the adoption and sustainability of CDSSs for antibiotic prescribing.

## Figures and Tables

**Figure 1 antibiotics-11-00384-f001:**
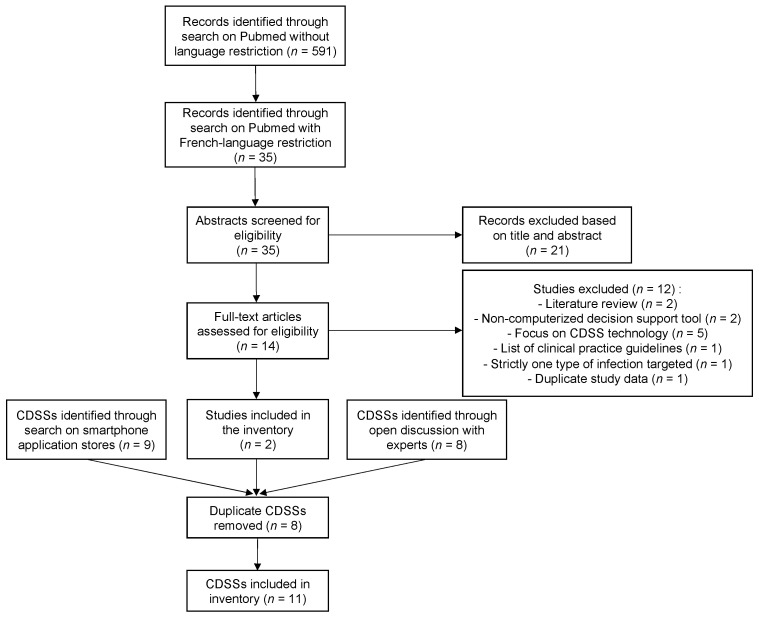
Flow chart of the CDSS selection process; Abbreviation: CDSS, clinical decision support system.

**Table 1 antibiotics-11-00384-t001:** Main characteristics of the CDSSs.

CDSS Characteristics	*n* (%)
**Source of funding**	
Public	8 (73)
Private	1 (9)
Unknown ^1^	2 (18)
**Access**	
**Fee-based**	3 (27)
Free of charge	8 (73)
**CDSS platform**	
Smartphone application	9 (82)
iOS	8 (73)
Android	8 (73)
Online website	8 (73)
Stand-alone software	2 (18)
Offline use	8 (73)
**Targeted users**	
Primary care physicians	11 (100)
Hospital physicians	9 (82)
**Targeted patients**	
Adults	10 (91)
Children	10 (91)
Pregnant women	10 (91)
Chronic kidney disease	5 (45)
Breastfeeding	2 (18)
**Targeted infections by the CDSS**	
Urinary tract infections	11 (100)
Genital infections	9 (82)
Upper respiratory tract infections	10 (91)
Lower respiratory tract infections	10 (91)
Skin and soft tissues infections	9 (82)
Digestive tract infections	10 (91)
Central nervous system infections	8 (73)
Cardiovascular infections	5 (45)
Bone and joint infections	8 (73)
Febrile neutropenia	8 (73)
Eye infections	5 (45)
CVC-related infections	4 (36)
Dental infections	5 (45)
Bloodstream infections	2 (18)
Prophylaxis	6 (55)
**Systematic Criteria for CDSS Decision ^2^**	
Site of infection	10 (91)
Nature of infection	11 (100)
Patient profile	2 (18)
**Types of information provided**	
**Types of Recommendation**	
Antibiotic selection	11 (100)
Priority ^3^	11 (100)
Allergy	11 (100)
Route of administration	11 (100)
Dose	10 (91)
Duration	10 (91)
GFR adaptation	5 (45)
Side effects	3 (27)
Locally adapted ^4^	3 (27)
Additional information provided	
Context and reminders ^5^	9 (82)
Scientific sources cited	9 (82)

Abbreviations: CVC, central venous catheter; GFR, glomerular filtration rate. ^1^ No information was found about the funding of the CDSS. ^2^ Mandatory information provided by users before accessing prescription recommendation. ^3^ Antibiotics are listed in preferred order. ^4^ Choice of antibiotics adapted to the local epidemiology. ^5^ Context and reminders included information about infection epidemiology, clinical presentation, diagnosis, and other treatment.

**Table 2 antibiotics-11-00384-t002:** Individual characteristics of the CDSSs regarding targeted users, patients, and infections.

CDSS Characteristics	Antibioclic	Antibiogarde	Antibiogilar	antibioGUIDE	Antibioguide	AntibioEst	APPLIBIOTIC	ePOPI	Prescriptor	Antibiothérapie Pédiatrique	AntibioHelp^®^
**Targeted users**											
Primary care physicians											
Hospital physicians											
**Targeted patients**											
Adults											
Children		 ^1^		 ^1^	 ^1^						
Pregnant women		 _1_	 _1_	 _1_	 _1_	 _1_	 _1_		 _1_		
Breastfeeding											
Chronic kidney disease											
**Targeted infections by the CDSS**											
Urinary tract infections											
Genital infections											
Upper respiratory tract infections											
Lower respiratory tract infections											
Skin and soft tissues infections											
Digestive tract infections											
Central nervous system infections											
Cardiovascular infections											
Bone and joint infections											
Febrile neutropenia											
Eye infections											
CVC-related infections											
Dental infections											
Bloodstream infections											
Prophylaxis											

Abbreviations: CDSS, clinical decision support system; CVC, central venous catheter. ^1^ Incomplete recommendation.

**Table 3 antibiotics-11-00384-t003:** Individual characteristics of the CDSSs regarding the types of information provided.

Types of Information Provided	Antibioclic	Antibiogarde	Antibiogilar	antibioGUIDE	Antibioguide	AntibioEst	APPLIBIOTIC	ePOPI	Prescriptor	Antibiothérapie Pédiatrique	AntibioHelp^®^
**Types of recommendation**											
Antibiotic selection											
Priority ^1^				 ^4^					 ^4^		
Allergy											
Route of administration			 ^4^	 ^4^							
Dose											
Duration											
GFR adaptation											
Side effects											
Locally adapted ^2^											
**Additional information provided**											
Context and reminders ^3^											
Scientific sources displayed											

Abbreviations: CDSS, clinical decision support system; GFR, glomerular filtration rate. ^1^ Antibiotics are listed in preferred order. ^2^ Choice of antibiotics adapted to the local epidemiology. ^3^ Context and reminders included information about infection epidemiology, clinical presentation, diagnosis, and other treatment. ^4^ Missing information.

## Data Availability

All data related to this study are available and accessible on request from the corresponding author.

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
