# Peer review of "Clinical Decision Support Systems for Antibiotic Prescribing: An Inventory of Current French Language Tools"

_antibiotics, 2022, doi:10.3390/antibiotics11030384_

Round 1

Reviewer 1 Report

This manuscript recognizes the inventory of French-language clinical decision support systems (CDSSs) used in community and hospital settings for an antibiotics prescription. They used literature reviews and open discussions from experts to identify commonly used decision support systems, including those that have not been published yet.

Strong points:

The literature search was thoroughly done and nicely presented in the manuscript. They have covered all the important published materials.

Weakness:

Although I do not have any major comments for improvement, the author can put more effort into elaborating and explaining Table 1.

Reviewer 2 Report

It is a good research word based on the global threat of AMR, antimicrobial stewardship programs (ASPs)and hence the topic of research is interesting to the readers of the journal Antibiotics. The manuscript follows the scope of the journal Antibiotics.

 I would recommend this manuscript could be published in Antibiotics with the following changes.

The authors need to address the below-mentioned queries.

1. The author needs to discuss table 1 and table 2 in the test to provide a more clear picture to the readers.

2. Tables 1 and 2 needs footnotes.

3. Table 2 could be divided into different parts based on CDSS characteristics and discuss details in the text.

4. The author needs to provide links for available CDSSs in the supplementary.

5. The author needs to discuss a bit the significant difference between CDSSs available for free vs required cost.

6. The author could comment if there is any preference or advantage of one CDSS over another based on infection for an antibiotic prescription.

7. The author could merge S1 and S2 into one document.

8. The conclusion needs to be more profound, and it lacks depth.

9. The author could include the following relevant reference.

1. Peiffer-Smadja N, Rawson TM, Ahmad R, Buchard A, Georgiou P, Lescure FX, Birgand G, Holmes AH. Machine learning for clinical decision support in infectious diseases: a narrative review of current applications. Clin Microbiol Infect. 2020 May;26(5):584-595. 

doi: 10.1016/j.cmi.2019.09.009. Epub 2019 Sep 17. Erratum in: Clin Microbiol Infect. 2020 Aug;26(8):1118. PMID: 31539636.

2, Crayton, E., Richardson, M., Fuller, C., Smith, C., Liu, S., Forbes, G., Anderson, N., Shallcross, L., Michie, S., Hayward, A., & Lorencatto, F. (2020). Interventions to improve appropriate antibiotic prescribing in long-term care facilities: a systematic review. BMC geriatrics20(1), 237. https://doi.org/10.1186/s12877-020-01564-1

Reviewer 3 Report

This manuscript is nice and generally well-written. The analyses and interpretations seem scientific and logical. However, it has to be improved for some aspects, as I suggest below:

Question 1: the authors described the study retrieval and selection processes in the results section. However, it would be interesting if they also report in the manuscript the flow-chart of study selection, based on specified inclusion/exclusion criteria and specifying the reasons for the exclusion. In addition, it would also be desirable for the reader to have an idea of the amount of published articles on the same subject, indipendently by CDSSs language, to be included in the start of the selection process; only after that the authors should select the French language CDSSs. Finally, is there any tool which is available in English or other language, too? 

Question 2: as the authors write in the Discussion “the CDSSs described are that most commonly used by clinicians”, I wonder how did the authors have established that they are the most used? Could it also depend on how long the tools are available on the market or if they are offered for free? The authors should discuss these aspects and if they may have any impact on the retrieved results.

Question 3: the article could be more interesting for the reader if the authors could add some comments on the effectiveness of retrieved French-language tools in improving an appropriate medical prescribing of antibiotics or a better management of antibiotic use by physicians or patients; the authors should discuss these aspects or specify if there is no evidence or no available studies supporting their use in clinical practice.  
